# Lewy body pathology exacerbates brain hypometabolism and cognitive decline in Alzheimer's disease

Lyduine E. Collij [1,2,3] ✉, Sophie E. Mastenbroek[1,2,3], Niklas Mattsson-Carlgren[1,4,5], Olof Strandberg[1], Ruben Smith [1,6], Shorena Janelidze[1], Sebastian Palmqvist [1,6], Rik Ossenkoppele [1,7,8] & Oskar Hansson [1,6] ✉

Identifying concomitant Lewy body (LB) pathology through seed amplification assays (SAA) might enhance the diagnostic and prognostic work-up of Alzheimer's disease (AD) in clinical practice and trials. This study examined whether LB pathology exacerbates AD-related disease progression in 795 cognitively impaired individuals (Mild Cognitive Impairment and dementia) from the longitudinal multi-center observational ADNI cohort. Participants were on average 75 years of age (SD = 7.89), 40.8% were female, 184 (23.1%) had no biomarker evidence of AD/LB pathology, 39 (4.9%) had isolated LB pathology (AD-LB+), 395 (49.7%) had only AD pathology (AD+LB-), and 177 (22.3%) had both pathologies (AD+LB+). The AD+LB+ group showed worst baseline performance for most cognitive outcomes and compared to the AD +LB− group faster global cognitive decline and more cortical hypometabolism, particularly in posterior brain regions. Neuropathological examination (n = 61) showed high sensitivity (26/27, 96.3%) and specificity (27/28, 96.4%) of the SAA-test. We showed that co-existing LB-positivity exacerbates cognitive decline and cortical brain hypometabolism in AD. In vivo LB pathology detection could enhance prognostic evaluations in clinical practice and could have implications for clinical AD trial design.

Lewy body (LB) pathology is the second most common proteinopathy among dementias and characterized by the intraneuronal aggregation of misfolded α-synuclein. While the accumulation of α-synuclein is the pathological hallmark of Lewy body diseases (e.g. Parkinson's disease and dementia with Lewy bodies [DLB]), LB pathology is also a common co-pathology in patients with Alzheimer's disease (AD). In fact, *post-mortem* studies showed that approximately 30-40% of sporadic AD patients exhibit comorbid α-synuclein pathology in addition to amyloid-β (Aβ) and tau depositions[1,2]. However, due to a lack of reliable biomarkers for LB pathology until recently, the clinical and biological effects of LB pathology on disease progression remain poorly understood.

The advent of cerebrospinal fluid (CSF) seed amplification assays (SAA) now enables reliable in vivo detection of α-synuclein

[1]Clinical memory Research Unit, Department of Clinical Sciences Malmö, Faculty of Medicine, Lund University, Lund, Sweden. [2]Radiology and Nuclear Medicine, Amsterdam UMC, location VUmc, Amsterdam, the Netherlands. [3]Brain Imaging, Amsterdam Neuroscience, Amsterdam, the Netherlands. [4]Department of Neurology, Skåne University Hospital, Lund University, Lund, Sweden. [5]Wallenberg Center for Molecular Medicine, Lund University, Lund, Sweden. [6]Memory Clinic, Skåne University Hospital, Malmö, Sweden. [7]Neurology, Alzheimercenter Amsterdam, Amsterdam UMC, location VUmc, Amsterdam, the Netherlands. [8]Neurodegeneration, Amsterdam Neuroscience, Amsterdam, the Netherlands. ✉e-mail: lyduine.collij@med.lu.se; oskar.hansson@med.lu.se

pathology[3,4], with high diagnostic accuracy demonstrated in both neuropathological[4-7] and clinical studies[8,9]. Recent results of a large heterogeneous memory clinic sample revealed that even when adjusting for AD pathology, the presence of LB pathology was associated with LB-specific clinical features such as worse attention/executive, visuospatial, and motor functioning, as well as an increased prevalence of hallucinations[10]. In addition, individuals with both AD and LB pathology ($n = 98$) exhibited the fastest cognitive deterioration, though not significantly different from those with only AD pathology ($n = 377$), which might have been due to lack of statistical power[10]. Consequently, larger longitudinal studies are needed to elucidate to what extent the presence of LB pathology exacerbates AD-related cognitive decline and associated regional neuronal dysfunction. Regarding the latter, [$^{18}$F]-fluorodeoxyglucose (FDG) PET can determine pathology-specific patterns of hypometabolism and is used in clinical practice to support the differential diagnosis of AD versus DLB[11]. Together, understanding how concomitant LB pathology affects cognition and neuronal dysfunction is important for how to integrate α-synuclein SAA tests in the diagnostic and prognostic work-up of individuals with cognitive impairment in both clinical practice and trials.

We therefore aimed to study the effects of LB pathology on longitudinal cognitive functioning and FDG-PET in cognitively impaired individuals enrolled in the Alzheimer's Disease Neuroimaging Initiative (ADNI) cohort ($N = 795$). Using *post-mortem* assessment of Aβ, tau, and LB pathology in a subset of 61 cases, we verified the accuracy of the α-synuclein SAA test to detect LB pathology.

## Results

At baseline, participants were on average 75 years of age ($SD = 7.89$), 40.8% were female, 71.9% were AD+, and 27.2% were LB+. The prevalence of AD/LB groups for the whole cohort, stratified by cognitive state, and the *post-mortem* population ($n = 61$) is shown in Fig. 1A, B, respectively. In the clinical population, 184 (23.1%) individuals were biomarker negative, 39 (4.9%) had isolated LB pathology, 395 (49.7%) had only AD pathology, and 177 (22.3%) had both pathologies. As expected, the group distributions were dependent on baseline cognitive status. Isolated LB pathology was uncommon independent of cognitive status, but the relative prevalence was higher in MCI (6.7%) compared to AD dementia (2.4%) (Table 1 and Supplementary Table 1).

The three biomarker positive groups were significantly older than the AD-LB- population ($F = 6.78$, $p < 0.001$) but did not differ from each other. AD+ populations were more often *APOE*-ε4 carriers than AD- populations, and participants with isolated LB pathology were least often *APOE*-ε4 carriers ($\chi^2 = 158.98$, $p < 0.001$). Though AD/LB groups were not significantly associated with sex ($p = 0.06$), there was a strong tendency for more males in the AD-LB+ group (79.5% vs. 57.1%-60.5%, Table 1). AD/LB groups were not predictive of presence of tremor, abnormal gait, or reduced motor strength (Table 1).

### AD/LB group status is associated with cognitive functioning

At baseline, the AD+LB- and AD+LB+ groups performed worse on the MMSE (AD+LB-: β = −1.13, SE = 0.30, $p = 0.001$; AD+LB+ : β = −2.05, SE = 0.36, $p < 0.0001$) and PACC (AD+LB-: β = −0.03, SE = 0.05, $p < 0.0001$; AD+LB+ : β = −0.05, SE = 0.06, $p < 0.0001$) compared to the AD-LB- group. Further, the AD+LB+ group showed worse baseline performance compared to the other groups for both the MMSE (vs AD-LB+ : β = −2.20, SE = 0.57, $p = 0.0007$; vs AD+LB-: β = −0.92, SE = 0.29, $p = 0.008$) and PACC (vs AD-LB+ : β = −0.05, SE = 0.09, $p < 0.0001$; vs AD+LB-: β = −0.03, SE = 0.05, $p = 0.019$, Fig. 2A, B).

Linear mixed models (LMMs) including an additional quadratic term for time (time$^2$) were preferred for the MMSE and PACC based on ΔBIC (Table 2). Over time, both the AD+ LB- and AD+LB+ groups showed a significant initial (time) and accelerated (time$^2$) decline compared to the AD- groups (Table 2). The accelerated decline

was more pronounced for the AD+LB+ group than the AD+LB- group (MMSE: $\beta_{AD/LB\ group*time}^2 = -0.15$, SE = 0.05, $p = 0.005$, PACC: $\beta_{AD/LB\ group*time}^2 = -0.026$, SE = 0.008, $p = 0.0015$, Supplementary Table 3), indicating that those with LB pathology in addition to AD pathology accelerated in global cognitive decline over time compared to those with isolated AD pathology (Fig. 2C, D). Results were consistent within the MCI population (Supplementary Tables 4, 5 and Supplementary Fig. 2A, B).

For the domain-specific cognitive composites, AD+LB- and AD+LB+ groups performed worse on most cognitive domains compared to the AD-LB- group at baseline and over time. In addition, the AD+LB+ group performed worse than the AD+LB− group for memory, executive functioning, and language, but not visuospatial functioning at baseline. Over time, no cognitive domain-specific differences between AD+LB− and AD+LB+ groups were observed. Results were similar for the MCI population, with the most apparent difference being that the isolated LB+ group performed worse on visuospatial functioning over time compared to both the AD-LB- and AD+LB- groups. Detailed results are described in Supplementary Results.

### AD/LB group status is associated with regional metabolism

In the cross-sectional analysis, both AD+LB- and AD+LB+ groups showed widespread cortical hypometabolism when compared to the AD-LB- group (Fig. 3A). Further, the AD+LB+ group demonstrated greater hypometabolism in mainly posterior regions compared to the AD+LB− group, including the superior, middle, and inferior occipital cortex, cuneus, angular cortex, superior and inferior lateral parietal cortex, precuneus, and middle temporal cortex (Fig. 3B). Combining these regions into a meta-ROI, we observed that the AD+LB+ group exhibited faster glucose hypometabolism than the AD+LB− group at a statistical trend level ($p = 0.06$, Fig. 3C). Finally, no significant effect of AD/LB groups on SN metabolism were observed (Supplementary Fig. 3).

### Post-mortem validation of the SAA-measure

Complete *post-mortem* assessments of Aβ, tau, and LB pathology were available for 61 individuals, with an average interval from SAA testing to death of 3.26 (2.20) years (Supplementary Table 6). Nearly all cases with an abnormal α-syn SAA test had LB pathology at autopsy (26/27, 96.3% sensitivity). The discrepant case was classified as 'other 4 R tauopathy'. Among the cases with cortical LBs indicative of widespread pathology, 90% (18/20) had a positive a-synuclein SAA test. In turn, only 63.6% (7/11) of those with relatively early LB pathology localized predominantly in the brainstem or amygdala had a positive SAA test. Among those with no LB pathology, 96.4% (27/28) had a negative SAA test result, illustrating the tests high specificity (Fig. 1B).

LB pathology was more pronounced in the AD+LB+ group compared to AD+LB-, with 69.6% (16/23) cases showing cortical depositions compared to 7.4% (2/27). All individuals with an isolated abnormal a-synuclein SAA test (AD-LB+) had confirmed LB pathology at post-mortem assessment across early and more advanced stages (Supplementary Table 6).

As expected, AD-related pathological scores (Thal, Braak, and CERAD) were all significantly higher in AD+ individuals compared to AD-, but the neuropathology AD scores did not significantly differ between AD+LB− and AD+LB+ groups (Fig. 1B). Finally, TDP-43 pathology in the hippocampus and amygdala did not differ between groups (Supplementary Table 7).

## Discussion

In this longitudinal study, we investigated the effect of LB co-pathology as measured with the α-synuclein CSF-SAA test, on AD disease progression in cognitively impaired individuals. We show that the presence of LB pathology exacerbates global cognitive decline in symptomatic AD and is associated with more pronounced

**A) Clinical population**

**Whole cohort** (*N*=795)

**Mild Cognitive Impairment** (*N*=466)

**Alzheimer's Dementia** (*N*=329)

**B)** *Post-mortem* population

Lewy body pathology

Thal stages (Amyloid plaques)

CERAD (neuritic plaques)

Braak stages

TDP-43 (hippocampus)

- AD+LB+
- AD+LB−
- AD−LB+
- AD−LB−

**Fig. 1 | Prevalence of CSF AD/LB groups in clinical and *post-mortem* populations. A** Barplots illustrate the prevalence of AD/LB groups across the whole cohort and split by MCI and dementia individuals. AD+ is defined as CSF p-tau$_{181}$/A$\beta_{42}$ positive and LB+ as α-synuclein SAA positive. **B** CSF AD/LB groups distribution within the *post-mortem* subpopulation is shown across neuropathological scores of regional distribution of Lewy body pathology, amyloid plaque Thal stages, neuritic plaques CERAD grading, tau Braak stages, and presence of TDP-43 in the hippocampus.

hypometabolism, particularly in posterior cortical regions. Finally, the accuracy of the CSF-SAA test was further corroborated by the *post-mortem* data. These findings have implications for clinical practice and clinical trial design, as in vivo detection of concomitant LB pathology could improve the prognostic work-up of AD.

Our results expands the recent work by Quadalti and colleagues (2023), who previously demonstrated that LB pathology affected the clinical profile of cognitively impaired individuals[10]. However, while they showed that individuals with both AD and LB pathology exhibited the fastest cognitive deterioration, this did not significantly differ from those with AD pathology only. In the present study, we implemented

LMMs with quadratic terms for time to account for previously described non-linear changes in cognitive decline[12]. We observed that patients with biomarker evidence for both AD and LB pathology specifically demonstrated an accelerated global cognitive decline as measured by the MMSE and PACC compared to the AD+LB− group. Importantly, our neuropathological results suggest that this is not merely due to more severe AD pathology in the mixed pathology group, but rather the additive effect of concomitant LB pathology. In line, recent work in the same cohort suggests that α-synuclein pathology was associated with poorer cognition particularly when tau was low[13]. In addition, and in line with previous *post-mortem*

**Table 1 | Baseline characteristics**

| | AD-LB-(N = 184) | AD-LB+ (N = 39) | AD+LB-(N = 395) | AD+LB+ (N = 177) | Overall (N = 795) |
|---|---|---|---|---|---|
| Demographics | | | | | |
| Cognitive state | | | | | |
| MCI | 168 (36.1%) | 31 (6.7%) | 200 (50.6%) | 67 (37.9%) | 466 (58.6%) |
| Alzheimer's dementia | 16 (4.9%) | 8 (2.4%) | 195 (49.4%) | 110 (62.1%) | 329 (41.4%) |
| Age | 72.8 (8.59) | 76.0 (8.74) | 75.2 (7.47) | 76.3 (7.44) | 74.9 (7.89) |
| Sex, F (%) | 79 (42.9%) | 8 (20.5%) | 167 (42.3%) | 70 (39.5%) | 324 (40.8%) |
| APOE-ε4 | | | | | |
| Non-carrier | 144 (78.3%) | 35 (89.7%) | 128 (32.4%) | 56 (31.6%) | 363 (45.7%) |
| Heterozygous | 40 (21.7%) | 3 (7.7%) | 196 (49.6%) | 85 (48.0%) | 324 (40.8%) |
| Homozygous | 0 (0%) | 1 (2.6%) | 71 (18.0%) | 36 (20.3%) | 108 (13.6%) |
| Neurological functioning[a] | | | | | |
| Tremor | 24 (13.0%) | 2 (5.1%) | 56 (14.2%) | 25 (14.1%) | 107 (13.5%) |
| Abnormal gait | 22 (12.0%) | 7 (17.9%) | 45 (11.4%) | 28 (15.8%) | 102 (12.8%) |
| Motor strength | 14 (7.6%) | 3 (7.7%) | 23 (5.8%) | 6 (3.4%) | 46 (5.8%) |
| Cognition[b] | | | | | |
| MMSE | 28.1 (2.05) | 27.5 (2.30) | 24.7 (4.13) | 23.1 (5.08) | 25.3 (4.32) |
| PACC | −0.42 (0.473) | −0.54 (0.52) | −1.17 (0.72) | −1.43 (0.78) | −1.02 (0.77) |
| Memory | 0.37 (0.633) | 0.18 (0.64) | −0.44 (0.67) | −0.68 (0.69) | −0.28 (0.77) |
| Language | 0.47 (0.546) | 0.49 (0.48) | 0.04 (0.66) | −0.24 (0.66) | 0.10 (0.69) |
| Executive functioning | 0.50 (0.549) | 0.30 (0.52) | −0.07 (0.76) | −0.40 (0.75) | 0.01 (0.77) |
| *Missing* | 0 (0%) | 0 (0%) | 1 (0.3%) | 1 (0.6%) | 2 (0.3%) |
| Visuospatial | −0.01 (0.352) | 0.04 (0.32) | −0.14 (0.55) | −0.33 (0.69) | −0.16 (0.57) |
| *Missing* | 91 (49.5%) | 20 (51.3%) | 116 (29.4%) | 36 (20.3%) | 263 (33.1%) |
| PET imaging | | | | | |
| FDG-PET[c] | 1.26 (0.14) | 1.22 (0.17) | 1.11 (0.16) | 1.05 (0.14) | 1.14 (0.17) |
| *Missing* | 19 (10.3%) | 5 (12.8%) | 63 (15.9%) | 34 (19.2%) | 121 (15.2%) |

*MCI* mild cognitive impairment, *MMSE* Mini-Mental State Examination, *PACC* Preclinical Alzheimer Cognitive Composite.
[a]Represent presence of tremor or abnormal gait/motor strength.
[b]Composite cognitive scores are in z-scores.
[c]Global AD ROI, as processed by the ADNI PET-core group.

findings[14], we also observed that the combined LB/AD pathology group performed worse on most cognitive tests compared to the isolated LB-positive group. Considering the non-negligible percentage of concomitant LB pathology in AD patients as demonstrated in recent (20.6%[10], 45.0%[15]) and the current work (30.9%), determining LB biomarker status is essential to inform on prognosis in a clinical setting and support risk-stratification and modeling efforts for AD clinical trials. If co-pathology is not considered, the effect of AD disease-modifying therapies could be more limited, as concomitant LB pathology could be driving disease progression despite successful removal of Aβ.

The exacerbated effect of concomitant LB pathology on disease progression was also apparent in terms of hypometabolism as measured by FDG-PET, a commonly used clinical tool for both AD and DLB[11]. Previous studies have described a distinct parieto-occipital hypometabolism in cognitively impaired patients with DLB[16,17]. However, due to the absence of LB pathology in vivo biomarkers, studies on hypometabolism patterns in mixed AD/LB pathology are lacking. A recent study in autopsy-confirmed pure-AD, AD-LB, and pure-LB patients suggested that patients with mixed pathology had a highly similar spatial pattern to pure-AD patients, reporting unexpectedly no significant differences between the two groups. However, when investigating a subpopulation with relatively high LB and low AD burden, the expected DLB-like hypometabolism pattern was observed[18]. Here, we provide evidence that cognitively impaired individuals with biomarker evidence of mixed pathology have additional hypometabolism to their pure-AD counterparts, in mostly parieto-

occipital regions commonly associated with LB pathology, but also in a key AD-associated region, namely the middle temporal cortex. This more apparent cortical neuronal dysfunction probably underlies the observed exacerbated cognitive decline of the AD+LB+ group in the current study. In contrast to previous work[16], we did not observe differences in metabolism of the SN between AD+LB- and AD+LB+ individuals, which is considered a marker of DLB-specific neurodegeneration[19]. This is probably due to the fact that ADNI is primarily an AD cohort, with a limited number of patients with parkinsonism.

Implementation of a novel biomarker in the clinical routine or trials requires excellent performance. Previous autopsy-based studies have demonstrated the very high diagnostic accuracy of the α-syn SAA test[4,20], which we further corroborate in the current study, showing ~96% specificity and sensitivity. In line with previous work[5,21], we demonstrate that the majority of individuals with cortical LB had a positive test result 90% (18/20), while for those with deposition in early regions (i.e., predominantly in the brainstem or amygdala), this was 63.6% (7/11). Interestingly, subjects with early LB deposition that were not classified as LB-positive based on the CSF α-syn SAA test were most often AD cases, which is in line with an abundance of literature most frequently describing LB co-pathology in the amygdala in AD[22–25].

The current work has some limitations that should be considered. First, the cohort included a relatively small number of isolated LB+ individuals (4.9%), which is even lower than what has been observed in cognitively unimpaired individuals[26]. This resulted in a

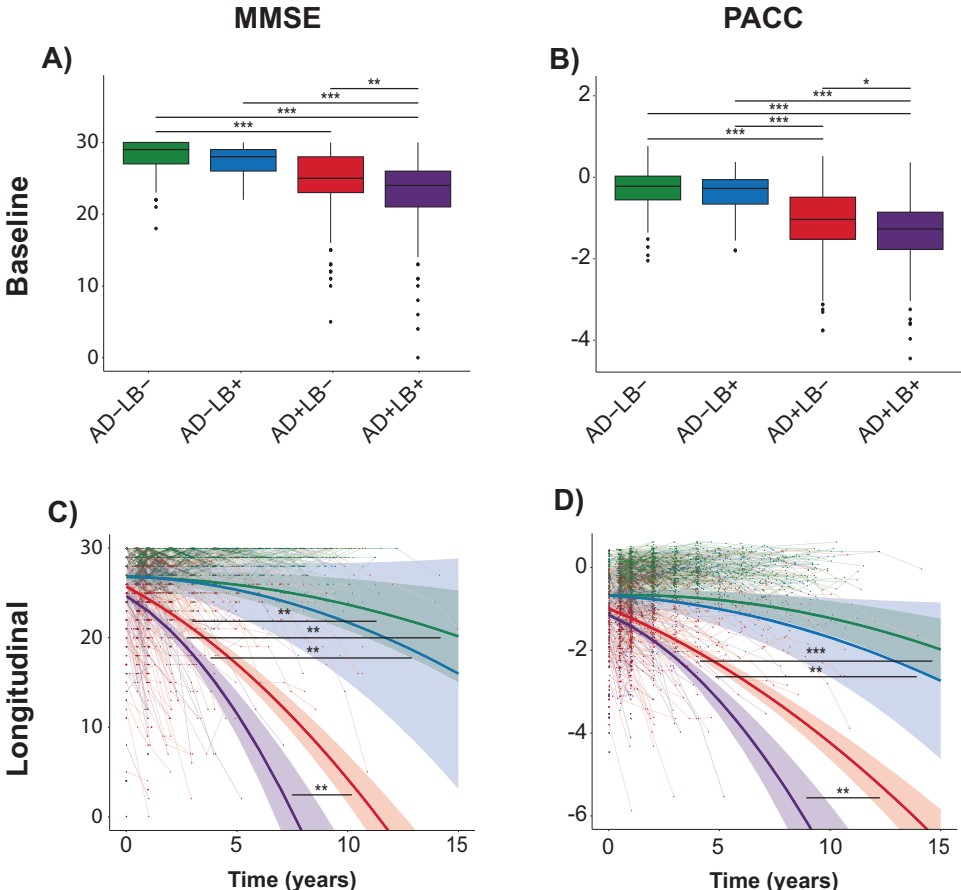

**Fig. 2 | AD/LB groups and global cognitive functioning.** Figure illustrates AD/LB group differences for (**A**, **B**) cross-sectional cognition based on the two-sided generalized linear model ($n = 795$) and (**C**, **D**) longitudinal cognitive performance based on the two-sided linear mixed models. All models were corrected for baseline age, sex, cognitive state, and level of education. Boxplots show the median, lower, and upper quartiles with whiskers representing minimum and maximum values.

The spaghetti plots illustrate raw data regarding cognitive performance over time, while lines represent model fits (shaded area reflect 95% confidence interval). Models including an additional quadratic term for time better described the data. Lines within figure represent significant differences in AD/LB group*time$^2$. Only p-values adjusted for multiple comparison are shown. *$p_{adjusted} < 0.05$, **$p_{adjusted} < 0.01$, ***$p_{adjusted} < 0.001$.

**Table 2 | Results linear mixed models cognition**

|  | AD-LB + *Time | AD + LB-*Time | AD + LB + *Time | AD-LB + *Time$^2$ | AD + LB-*Time$^2$ | AD + LB + *Time$^2$ | ΔBIC |
|---|---|---|---|---|---|---|---|
| MMSE<br>N = 795, 3171 obs | −0.06 (0.34)<br>0.86 | −1.28 (0.18)<br><6.29e-12 | −1.43 (0.24)<br><2.38e-09 | −0.02 (0.03)<br>0.64 | −0.06 (0.02)<br>0.0004 | −0.21 (0.05)<br>0.0001 | −39<br><1.027e-14 |
| PACC<br>N = 795, 3167 obs | −0.04 (0.05)<br>0.42 | −0.23 (0.03)<br><2e-16 | −0.24 (0.03)<br><2.67e-12 | −0.01 (0.05)<br>0.42 | −0.004 (0.002)<br>0.09 | −0.03 (0.01)<br>0.0002 | −36<br><5.211e-14 |

β (SE), p-value vs. reference group (AD-LB-); ΔBIC: difference in BIC between two-sided linear and quadratic model. A negative value indicates preference for the quadratic model, while a positive term indicates preference for the linear model. P-value is from the model comparison chi-square statistic. Significant interactions between AD/LB group and time or time$^2$ reflect initial and accelerated decline in cognitive functioning, respectively. P-values are adjusted for multiple comparisons.
*MMSE* Mini-Mental State Examination, *PACC* Preclinical Alzheimer Cognitive Composite.

limited statistical power to assess effects on disease progression of this group. One explanation could be the ADNI exclusion criteria regarding Parkinsonian symptoms and neurological dysfunction, as the ADNI cohort was designed to resemble an AD clinical trial population. Nonetheless, longitudinal results regarding cognitive domains functioning suggests that isolated LB-positivity is not benign. Second, the above mentioned exclusion criteria could also explain the lack of an association between LB pathology and features of parkinsonism (Table 1), which is in contrast to previous work in both cognitively unimpaired[26] and impaired[10] populations. In addition, these features of parkinsonism and other more LB-pathology associated symptoms such as visual hallucinations might not have

been optimally measured in this multicenter study. Third, the number of subjects with longitudinal FDG-PET and associated follow-up time was limited, which could have reduced the power to detect an effect of additional LB pathology on hypometabolism over time. Fourth, no imaging of the dopaminergic system through for example DAT-SPECT was available, which is also of clinical interest considering that this imaging technique is a core tool for the diagnosis of DLB[11]. Finally, due to the lack of a valid and reliable in-vivo biomarker for TDP-43, this common co-pathology to AD and mixed AD/DLB cases[27] was not included in the current work. In line, our post-mortem analysis demonstrated that comorbid TDP-43 pathology (amygdala) was observed in approximately 35% of AD+ cases,

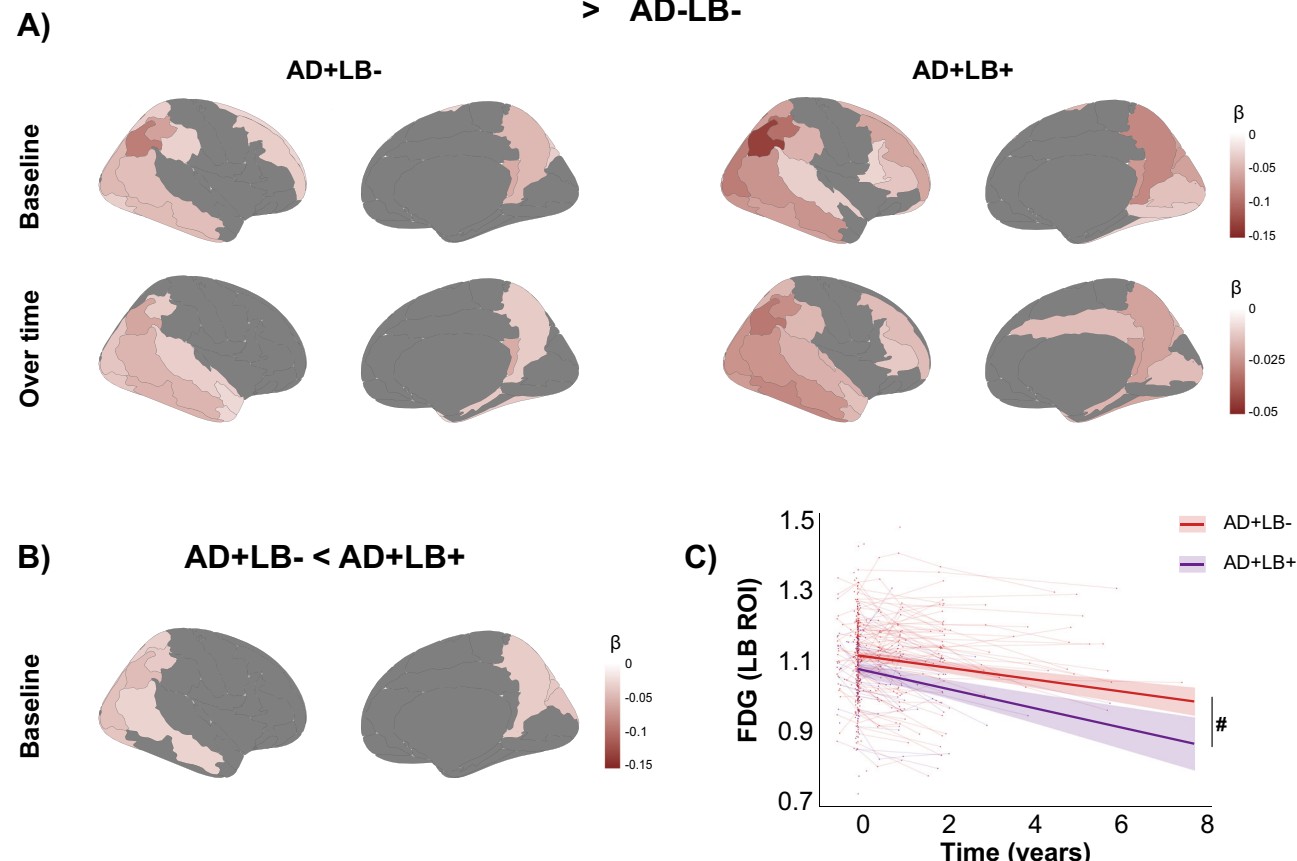

**Fig. 3 | Regional hypometabolism as measured with FDG-PET.** Figure illustrates group differences assessed with two-sided linear mixed models in regional metabolism as measured with FDG-PET between AD/LB groups in the CI population ($n = 568$), after correction for baseline age, sex, and cognitive state. Only results that survived FDR-correction are shown. **A** Effects of AD/LB pathological groups vs the AD-LB- reference group, while (**B**) illustrates regions for which the AD+LB+ group demonstrated more severe hypometabolism at baseline compared to the AD+LB− group. **C** The spaghetti plots illustrate raw data regarding FDG-PET over time, while lines represent model fits (shaded area reflect 95% confidence interval), demonstrating a non-significant trend towards more apparent hypometabolism over time in the meta-ROI, consisting of regions identified in (**B**). #$p = 0.06$.

respectively. TDP-43 co-pathology has been associated with greater cognitive decline[28,29] and therefore of great interest for future studies.

# Methods
## Study cohort
Data was retrieved from the ADNI database. The ADNI study was launched in 2003 as a public-private partnership led by principal investigator Michael W. Weiner, MD (adni.loni.usc.edu/). For ADNI, the study was approved after ethical review of each site's local review board and all participants provided informed written consent (ClinicalTrials.gov registry numbers: ADNI GO: NCT01078636; ADNI 1: NCT00106899; ADNI 2: NCT01231971). As per ADNI protocols, all procedures performed in studies involving human participants were in accordance with the ethical standards of the institutional and/or national research committee and with the 1964 Helsinki Declaration and its later amendments or comparable ethical standards.

We selected all cognitively impaired participants for which α-synuclein status was determined using the CSF α-synuclein SAA test (using the last available CSF time-point) and who had available data on CSF Aβ$_{42}$ and p-tau$_{181}$ within one year of α-synuclein SAA status. Cases with an intermediate SAA result were excluded ($n = 17$). Of note, ADNI exclusion criteria includes overt Parkinsonian symptoms and neurological dysfunction. As such, patients with PD or DLB are not included in the dataset, which is therefore not representative of such patient populations.

## Biomarkers of Aβ, tau, and α-synuclein
CSF concentrations of Aβ$_{42}$ and p-tau$_{181}$ were measured using the Elecsys CSF immunoassay[30]. The CSF p-tau$_{181}$/Aβ$_{42}$ ratio was used to define AD-positivity based on a previously specified threshold of 0.021 against [$^{18}$F]florbetapir amyloid-PET, as recommended by the ADNI Biomarker Core Steering Committee (https://adni.loni.usc.edu/methods/).

To determine LB status, the α-synuclein seed amplification assay (SAA) was performed in the Amprion Clinical Laboratory (CLIA ID No. 05D2209417; CAP No. 8168002) using a method validated for clinical use in accordance with Clinical Laboratory Improvement Amendment (CLIA) requirements. A detailed description of the method can be found in Arnold et al., (2022)[7]. Participants were classified into one of four pathology groups based on their p-tau$_{181}$/Aβ$_{42}$ ratio and SAA α-syn status; i.e., "AD-LB-", "AD-LB+", "AD+LB-" or "AD+LB+".

## Neurological and neuropsychological assessments
All participants had available binary neurological assessments (i.e. normal or abnormal) regarding tremor, gait, and motor strength and completed a neuropsychological test battery at baseline (i.e., within 1 year of SAA), including the Mini-Mental State Examination (MMSE). Baseline composite scores were available for most subjects, including the Preclinical Alzheimer Cognitive Composite[31,32] ($n = 795$, [PACC]) and cognitive domain scores[33], reflecting memory ($N = 795$), language ($n = 795$), executive function ($n = 793$), and visuospatial function ($n = 532$). Longitudinal assessments were mostly available for the

MMSE and PACC (710 [89.3%] subjects with a mean follow-up time of 3.27 years, range = 0.5–15.9) and least available for the visuospatial domain (532 [66.9%] subjects with a mean follow-up time of 2.7 years, range = 0.5–12.2). Detailed longitudinal numbers per pathological group and cognitive outcomes can be found in Supplementary Table 7.

### FDG-PET acquisition and processing
Brain metabolism was investigated using FDG-PET acquisition 30–60 min post-injection of 185 MBq [18F]FDG[34]. Pre-processed images were downloaded from the ADNI-LONI database. Regional standard uptake value ratios (SUVr) against the whole cerebellum reference region were extracted using the Automated Anaotmical Labeling (AAL) atlas at Lund University. Cortical values were averaged across the hemispheres, as more pronounced bilateral parieto-occipital hypometabolism has been reported for DLB patients[11,16]. Finally, changes in substantia nigra (SN) metabolism were assessed using the DISTAL (Deep Brain Stimulation Intrinsic Template) atlas[35], which has been shown to be of added value in distinguishing prodromal AD from prodromal DLB patients[16]. In total, 568 participants had available FDG-PET within one year of the SAA measurement, which was considered their baseline assessment. Additionally, 222 participants underwent longitudinal FDG-PET imaging, with an average follow-up of 2.5 years (range = 0.5–9.2 years).

### Post-mortem assessments
All neuropathologic assessments were performed by the same neuropathologist (Dr. Nigel Cairns) at the central laboratory of the ADNI Neuropathology Core at the Knight Alzheimer's Disease Research Center, which provides uniform neuropathologic assessments of deceased ADNI participants[36]. Evidence of LB pathology was assessed according to modified McKeith criteria (no, brainstem/amygdala predominant, cortical, and olfactory bulb)[37,38]. AD pathological scores were acquired according to the National Institute on Aging-Alzheimer's Association guidelines[37]. Three rating scales are used to describe core hallmarks of AD neuropathology, including Thal phase for the location of Aβ plaques (ranging from 0 to 5), Braak stages for the location of tau neurofibrillary tangles (NFT) pathology (ranging from 0 to 6), and Consortium to Establish a Registry for Alzheimer's Disease (CERAD) scores for density of neuritic plaques (ranging from 0 to 3).

### Statistical analysis
No statistical method was used to predetermine sample size. Demographic differences between pathology groups (AD-LB−, AD+LB−, AD-LB+, AD+LB+) were determined using chi-square and ANOVA tests, as appropriate. Cross-sectional group differences in neurological outcomes and cognitive performance were assessed using logistic regression models and general linear models, respectively, corrected for age, sex, baseline cognitive state (i.e. MCI or dementia), and (for models with cognition as outcome) level of education.

Linear mixed models (LMMs, R package lme4) with random slopes and intercepts were fitted to investigate the effect of pathology group on cognitive functioning over time, using an interaction term for group by time as predictor. All models were corrected for age, sex, baseline cognitive state, and level of education. To investigate potential non-linear changes, models additionally included an interaction between AD/LB group and time squared (time$^2$). A significant AD/LB group*time interaction is indicative of a group-dependent overall faster progression in cognitive decline, while a significant AD/LB group*time$^2$ was indicative for accelerated decline in cognitive functioning. Model preference (i.e., linear vs quadratic model) was determined using the Bayesian Information Criterion (BIC)[39], where the model with a lower BIC was selected. Pair-wise comparisons between groups were corrected for multiple testing ($p_{FDR} < 0.05$, R package emmeans). As a sensitivity analysis, LMMs were repeated within the MCI population (N = 466) only.

LMMs with random slopes and intercepts were also fitted to investigate the effect of AD/LB group and its interaction with time on regional metabolism based on the AAL atlas. Regions that were significantly different at baseline between AD+LB− and AD+LB+ groups were merged into a meta-ROI and differences in longitudinal change between AD+LB− and AD+LB+ groups within this meta-ROI were assessed. Finally, changes in SN metabolism were investigated. All models were corrected for age, sex, and baseline cognitive state. Regional FDG-PET analyses were corrected for multiple testing ($p_{FDR} < 0.05$).

Finally, differences between AD/LB groups in pathological scores for LB pathology, Aβ plaques, NFT pathology, neuritic plaques, and presence of TDP-43 in the hippocampus were assessed using ordinal logistic regression models (R package MASS), corrected for sex, age at death, measurement interval (time difference between α-syn SAA measure and date of death) and post-mortem interval (PMI; time difference between death and autopsy).

All analyses were performed in R Studio (version 4.2.2). Significance was set at two-sided P < 0.05 and corrected for multiple comparison when applicable.

### Reporting summary
Further information on research design is available in the Nature Portfolio Reporting Summary linked to this article.

## Data availability
Data was retrieved from the ADNI-LONI database (adni.loni.usc.edu). The ADNI study was launched in 2003 as a public-private partnership led by principal investigator Michael W. Weiner, MD. The primary goal of ADNI has been to test whether serial magnetic resonance imaging (MRI), positron emission tomography (PET), other biological markers, and clinical and neuropsychological assessment can be combined to measure the progression of mild cognitive impairment (MCI) and early Alzheimer's disease (AD). For up-to-date information, see www.adni-info.org. The patient-level original and preprocessed data cannot be directly shared due to restrictions set by the consortium. Source data are provided with this paper.

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

## Acknowledgements

We acknowledge all members of the Alzheimer's Disease Neuroimaging Initiative. Work at the authors' research center was supported by Swedish Research Council (2018-02052, 2021-02219, and 2022-00775), ERA PerMed (ERAPERMED2021-184), the Knut and Alice Wallenberg foundation (2022-0231), the Strategic Research Area MultiPark (Multidisciplinary Research in Parkinson's disease) at Lund University, the Swedish Alzheimer Foundation (AF-980907, AF-994229), the Swedish Brain Foundation (FO2021-0293, FO2022-0204, and FO2023-0163), The Parkinson foundation of Sweden (1412/22), the Cure Alzheimer's fund, the Konung Gustaf V:s och Drottning Victorias Frimurarestiftelse, the Skåne University Hospital Foundation (2020-O000028), Regionalt Forskningsstöd (2022-1259 and 2022-1346) and the Swedish federal government under the ALF agreement (2022-Projekt0080, 2022-Projekt0107). The funding sources had no role in the design and conduct of the study; in the collection, analysis, interpretation of the data; or in the preparation, review, or approval of the manuscript.

## Author contributions

L.E.C. performed the statistical analyses and drafted the manuscript; S.E.M. prepared the data and provided feedback on the manuscript; N.M.C. provided feedback on the manuscript; O.S. processed the FDG-PET scans; R.S. provided feedback on the manuscript; S.J. provided feedback on the manuscript; S.P. provided feedback on the manuscript; R.O. supervised the work and provided feedback on the manuscript; O.H. supervised the work and provided feedback on the manuscript.

## Funding

## Competing interests

L.E.C. has acquired research support from GE Healthcare and Springer Healthcare (paid by Eli Lilly), both paid to institution. Dr. Collij's salary is supported by the MSCA Postdoctoral fellowship (#101108819) and Alzheimer Association Research Fellowship (#23AARF-1029663) grants. O.H. has acquired research support (for the institution) from AVID Radiopharmaceuticals, Biogen, Eli Lilly, Eisai, Fujirebio, GE Healthcare, and Roche. In the past 2 years, he has received consultancy/speaker fees from AC Immune, Alzpath, BioArctic, Biogen, Bristol Meyer Squibb, Cerveau,

Eisai, Eli Lilly, Fujirebio, Merck, Novartis, Novo Nordisk, Roche, Sanofi and Siemens. R.O. has received research funding from European Research Council, ZonMw, NWO, National Institute of Health, Alzheimer Association, Alzheimer Nederland, Stichting Dioraphte, Cure Alzheimer's fund, Health Holland, ERA PerMed, Alzheimerfonden, Hjarnfonden (all paid to the institutions). R.O. has received research support from Avid Radiopharmaceuticals, Janssen Research & Development, Roche, Quanterix and Optina Diagnostics, and has given lectures in symposia sponsored by GE Healthcare. He is an advisory board member for Asceneuron and Bristol Myers Squibb. All the aforementioned has been paid to the institutions. He is an editorial board member of Alzheimer's Research & Therapy and the European Journal of Nuclear Medicine and Molecular Imaging. S.P. has acquired research support (for the institution) from ki elements / ADDF and Avid. In the past 2 years, he has received consultancy/speaker fees from Bioartic, Biogen, Esai, Lilly, and Roche. N.M.C. has received funding from WASP and DDLS Joint call for research projects (WASP/DDLS22-066), EU Join Program Neurodegenerative Diseases (2019-03401). The remaining authors declare no competing interests.
