## [Peer Review File · Nature Communications]

Lewy body pathology exacerbates brain hypometabolism and cognitive decline in Alzheimer's diseaseREVIEWER COMMENTS

Reviewer #1 (Remarks to the Author):

This is a well-written, solid and impactful study providing important information regarding the role of Lewy body (LB) co-pathology in Alzheimer's disease (AD) cognitive decline. Co-pathologies in AD patients have great implications for diagnostic and prognostic work-up in clinical practice and trials. The current study utilized the recently developed and validated aSyn-SAA as a biomarker for in vivo LB pathology to analyze the contribution of LB pathology to longitudinal declines of cognitive functions and neuronal hypometabolism in an ADNI cohort. This study follows up on a recent article published by some of the same authors (ref 10, <https://pubmed.ncbi.nlm.nih.gov/37464058/>). The current study found significantly accelerated cognitive declines in AD+LB+ patients compared to the AD+LB- group. Furthermore, the data strengthened the validity of aSyn-SAA as an excellent in vivo biomarker of LB pathology by comparing the SAA results to neuropathological changes in deceased patients from the same cohort. They also confirmed the usage of FDG-PET as a differential diagnostic tool for AD and DLB.

The study is very strong and does not have very significant weaknesses. Perhaps one point not discussed that could be a weakness is the fact that similar to LB pathology, TDP-43 co-pathology has also been reported to be frequent (50-60%) in AD patients and associated with greater cognitive declines (<https://pubmed.ncbi.nlm.nih.gov/24659241/>; <https://pubmed.ncbi.nlm.nih.gov/26224156/>). It was also reported that the TDP-43 pathology could be found in about 53% of post-mortem brains with AD/LB co-pathologies (<https://pubmed.ncbi.nlm.nih.gov/27495267/>). Due to the lack of a validated and reliable in vivo biomarker for TDP-43 pathology, it is not possible at this time to assess whether part of the changes observed in this study could be attributed in part to TDP-43 co-pathology in AD patients. However, I recommend the authors to include TDP-43 staining in the samples with tissue available to analyze whether TDP-43 co-pathology is seen with similar or higher frequency in AD+LB+ and AD+LB- patients. At the very least, they should include as a limitation of this study the putative effect that other co-pathologies (including TDP-43, but maybe also vascular changes) could have in their results.

The authors describe that participants with isolated LB pathology were least often APOE- ϵ 4 carriers ($\chi^2=158.98$, $p<0.001$). This is somewhat surprising, considering previous articles showing that the APOE ϵ 4 allele has been associated with an increased risk of DLB and PDD (e.g. <https://www.neurology.org/doi/10.1212/WNL.0000000000006212>). Do the authors have an explanation for these findings?

Reviewer #2 (Remarks to the Author):

Reviewer #3 (Remarks to the Author):

As the authors correctly state, whilst co-pathology with LB in Alzheimer's disease has been well recognised and well studied in neuropathology studies, the lack of a reliable in vivo biomarker for LB pathology has limited addressing this question in clinical cohorts. Seed amplification assays have recently emerged and can finally answer these questions.

The key message of this study is that in those with mild cognitive impairment or Alzheimer's disease, the presence of Lewy body (LB) co-pathology as measured by seed amplification assays (SAA) is associated with worse cognitive performance cross sectionally, and longitudinal cognitive decline. Secondly, they also show increased cortical hypometabolism in posterior brain regions in those with LB co-pathology with FDG PET, and validate the SAA assay in a subgroup with postmortem brain tissue.

The significance of this study is twofold. Firstly, the last two years have seen the first positive clinical trials of disease modifying drugs in mild cognitive impairment (MCI) and early Alzheimer's disease (AD) (lecanemab¹ and donanemab²). These treatments both target amyloid pathology, lecanemab being approved by the FDA and donanemab under assessment. It is clear that whilst these drugs show an effect, there are limitations in terms of effect size and safety. This leads to a timely question of other factors and pathologies underlying disease progression in MCI/AD, and the authors quite rightly state that this could inform stratification for future clinical trials.

Secondly, this study has been made possible by a novel technology to detect Lewy body pathology in vivo in CSF, the SAA. Prior to this studies relied either on post mortem³, α -synuclein measurement in CSF, or using FDG as a proxy for presumed LB pathology⁴. Postmortem studies are thus limited to small sample sizes, FDG PET relies on the assumption of the pattern of hypometabolism being indicative of pathology, and CSF α -synuclein is not a reliable biomarker of LB pathology⁵. With SAA, the idea of measuring the impact of LB-copathology on clinical and imaging metrics has been made possible in a much larger sample size. It also follows that this could be practically carried out either in trials or a clinical setting. The findings are in line with the previous studies using postmortem or other measures, but the demonstration of this effect in a large cohort with a new technology is highly significant and novel, and could have direct implications for clinical trials.

Overall, the data are valid and robust. Firstly, the use of the ADNI cohort, which is a large, well phenotyped cohort, with detailed clinical, biomarker, and imaging data. The novel SAA has been well validated in Lewy body dementias, and detection of LB pathology in other dementias, but the

authors take the extra step of demonstrating the validity of the assay within this cohort in the subgroup with postmortem tissue. This identifies similar results to previously published work, suggesting a successful application of this technology to this cohort, and its validity as a marker of LB co-pathology in the larger cohort.

A greater understanding of other features that underly progression in AD beyond amyloid and tau, that can be measured by an in vivo biomarker, could certainly inform future clinical trials that may wish to stratify, test post-hoc, exclude from amyloid targeting trials, or recruit specifically to trials targeting synuclein. LB pathology is commonly found postmortem, but under researched, possibly due to the lack of in vivo biomarkers until recent years. For that reason, I would say this study represents a significant paper for the field. It would also be of interest in the closely related field of dementia with Lewy bodies, and in Parkinson's disease research.

To my knowledge, there are two other studies that explore clinical measures in relation to LB pathology in large neurodegeneration cohorts, including one in ADNI. The recent study in the Biofinder cohort⁶ included 883 individuals, but across diagnosis including LBD. This study identified that the presence of LB pathology, as measured by SAA was associated with attention/executive impairments, visuospatial, motor function, and hallucinations. This study identified more rapid progression independent of diagnostic group.

This submitted paper extends on that, as the AD+/LB- group had 377 people, and the AD+/LB+ group is 98 people. This new study has a similar sized AD group (395), but 177 with AD+/LB+ (dual pathology). This study using the ADNI cohort also differs in that those recruited met criteria for either MCI or AD, so whilst the previous study statistically controlled for the impact of LBD diagnosis on cognitive decline, this study includes larger numbers and only those with an initial AD or MCI diagnosis. This study, as the authors reference, did not detect a significant difference between the AD+ group and the AD+LB+ group in rate of cognitive decline, possibly due to lack a power, which has been solved by this paper.

Secondly, the study in the ADNI cohort⁷ showed that LB co-pathology as measured by SAA was associated with cross sectional cognitive impairment in individuals with low tau. This study however did not include data on longitudinal cognitive decline, the key finding of the submitted paper, and whilst included FDG PET did not look at how LB co-pathology impacted FDG PET.

The cohort is a large, well established cohort with rich multimodal data. The choice of CSF results for AD pathology and LB pathology within a year is a reasonable cut off. The standards for AD biomarker positivity are the standards recommended by the central ADNI team, which is appropriate. There is some missing cognitive data, particularly in visuospatial function, with less available at baseline and follow up. Despite this, it remains a large data set and this missingness does not impact the overall conclusions made by the paper.

I have no expertise in the specific FDG PET acquisition and processing, however the methods used are standardised within ADNI. Averaging cortical values across hemispheres is commonly performed and a reasonable assumption based on previous data that bilateral hypometabolism would be found.

Similarly, the post mortem assessments are following standard ADNI protocols. The McKeith criteria is a widely used and accepted standard for Lewy body pathology included in international

guidelines.

The statistical methods for demographic differences are standard. Using linear mixed effects models with an interaction term between group and time is a commonly used and accepted approach for longitudinal data. They included both random intercepts and slopes, and tested whether a nonlinear model better fit the data, and corrected p values for multiple testing appropriately. Sensible covariates are also included in the model.

It is notable the difference between AD+ and AD+LB+ was only significant in the model with the quadratic term, however this model is by far the preferred by BIC, so is the crucial one.

The imaging findings are appropriately corrected for multiple comparisons. Selecting the most different regions at baseline, merging into a meta ROI and then testing for longitudinal associations is reasonable. They also report appropriate correcting for multiple testing and inclusion of covariates.

They report the differences in baseline demographics, and age and sex and subsequently included as covariates in analysis.

Suggestions for improvement:

The authors have some interesting thoughts regarding the lack of associations between LB pathology and motor symptoms in this cohort, as ADNI specifically recruited those meeting criteria for AD, or MCI. It would be very interesting to see if this extends to the presence of visual hallucinations. Previous autopsy studies⁸ have highlighted that visual hallucinations are probably the best predictor of LB pathology in mixed AD/LB. Within ADNI itself, visual hallucinations⁹ were associated with similar FDG PET findings found with LB pathology in this paper. It would certainly be of interest clinically whether the novel application of SAAs supported this association with visual hallucinations in this larger cohort. It may also raise the question as to whether the groups with mixed LB and AD pathology, occipital hypoperfusion, and the possible presence of a core feature of DLB could be argued to have DLB with AD co-pathology. This may strengthen the authors argument of the usefulness of SAA in clinical practice or trials, to properly stratify and target interventions. (The ability to test this may rely on specific data requests/access/sharing with the ADNI cohort, and I would say the novelty and significance of the study are clear without this analyses, however it would improve things).

Some have argued that co-incident LB pathology do not impact cognitive decline at the MCI stage¹⁰. It may be of interest if there are sufficient numbers within those with an MCI and AD diagnosis subgroups, whether the effect on cognitive decline is the same or distinct at each disease stage.

It would be good if there was more explanation of the statistical method of pairwise comparison that led to the results in figure 2c + d, and described in supplementary table 3c. This may well be a standard approach, or I may have missed it in the methods, but it would help to fully understand this table as it contains what I read as the most important findings of the study.

It is also notable that there are no differences in any individual cognitive domain, although some of these in the table are listed as not available for the quadratic model, whilst not significant in the non quadratic model. Is this because of smaller subsets of data for these domains, and the model not working?

The changes in regional metabolism between AD+ and AD+LB+ groups are well described and the figures present this clearly. The longitudinal change is reported to be at the statistical trend level of 0.06. The limitations of p-values and arbitrary nature of a 0.05 cut off present limitations, but for better or worse it remains a standard. Whilst in the results section and figure this is presented, the figure description puts this as 'more apparent hypometabolism'. It might be simpler to describe this as a non significant trend towards longitudinal reductions in metabolism or similar.

Line 224 may need a bit more discussion/clarification – would describing the pathology as more widespread or contrasting the cortical with brainstem/amygdala help? It is clear what is meant from looking at the supplementary table but I think a little more in the text would be useful.

I believe the manuscript references the previous literature appropriately. It may be worth mentioning the Landau reference⁷ as it uses the SAA in the same cohort, although may well have been published after submission.

1. Van Dyck, C. H. et al. Lecanemab in Early Alzheimer's Disease. *N. Engl. J. Med.* 388, 9–21 (2023).
2. Sims, J. R. et al. Donanemab in Early Symptomatic Alzheimer Disease: The TRAILBLAZER-ALZ 2 Randomized Clinical Trial. *JAMA* 330, 512 (2023).
3. Toledo, J. B. et al. Clinical and multimodal biomarker correlates of ADNI neuropathological findings. *Acta Neuropathol. Commun.* 1, 65 (2013).
4. Silva-Rodríguez, J. et al. Characteristics of amnesic patients with hypometabolism patterns suggestive of Lewy body pathology. *Brain* 146, 4520–4531 (2023).
5. Gao, L. et al. Cerebrospinal fluid alpha-synuclein as a biomarker for Parkinson's disease diagnosis: a systematic review and meta-analysis. *Int. J. Neurosci.* 125, 645–654 (2015).
6. Quadalti, C. et al. Clinical effects of Lewy body pathology in cognitively impaired individuals. *Nat. Med.* 29, 1964–1970 (2023).
7. Landau, S. M. et al. Individuals with Alzheimer's disease and low tau burden: Characteristics and implications. *Alzheimers Dement.* 20, 2113–2127 (2024).
8. Thomas, A. J. et al. Improving the identification of dementia with Lewy bodies in the context of an Alzheimer's-type dementia. *Alzheimers Res. Ther.* 10, 27 (2018).
9. for the Alzheimer's Disease Neuroimaging Initiative, Pezzoli, S., Manca, R., Cagnin, A. & Venneri, A. A Multimodal Neuroimaging and Neuropsychological Study of Visual Hallucinations in Alzheimer's Disease. *J. Alzheimers Dis.* 89, 133–149 (2022).
10. Shriram, J., Malek-Ahmadi, M., Irwin, C. & Sabbagh, M. Impact of incidental synucleinopathy in mild cognitive impairment due to Alzheimer disease. *J. Neuropathol. Exp. Neurol.* 83, 230–237 (2024).

Reviewer #4 (Remarks to the Author):

This is a study of the additional detrimental effects of α -synuclein pathology in a large well-established cohort of mild cognitive impaired and demented individuals with presumed Alzheimer's disease. The methods are relevant and clearly described, figures easy to understand and results interpreted with respect to limitations. Overall the study is highly relevant and provide original and new insight to a major research field in neuroscience using a rather new but established method for measuring α -synuclein in CSF. In a subset of the patients, the method is further validated to post-mortem assessments.

I have only encountered a few issues that should be dealt with.

In the methods section, it says that participants with a CSF α -synuclein SAA test were selected. As this measure is not part of the standard test for ADNI subjects according to adni.loni.usc.edu, please describe in more detail how the subjects were selected. From which centers were they included and which inclusion criteria? Were any CSF α -synuclein SAA test performed retrospectively on biobank material? And if so how were the subjects selected? Detailed description of the selection of the included subjects is mandatory for evaluating the transferability of results to other patient groups.

40.8% were female. This seems like a low fraction. Do you have any explanation for this?

Patients are included as MCI or AD. Thus, patients with known LBD or PD will be excluded and LB pathology will be a random finding. Therefore, the AD-LB+ group is not representative for patients with symptoms from LB pathology. This limitation is stated briefly in the discussion but could be included in the description of the study cohort as well.

The three figures are nicely performed but I suggest to include of number of individuals in each group at baseline and number of individuals with follow-up data in the legends or figures to increase the readability.

REVIEWER COMMENTS

Reviewer #1 (Remarks to the Author):

This is a well-written, solid and impactful study providing important information regarding the role of Lewy body (LB) co-pathology in Alzheimer's disease (AD) cognitive decline. Co-pathologies in AD patients have great implications for diagnostic and prognostic work-up in clinical practice and trials. The current study utilized the recently developed and validated aSyn-SAA as a biomarker for in vivo LB pathology to analyze the contribution of LB pathology to longitudinal declines of cognitive functions and neuronal hypometabolism in an ADNI cohort. This study follows up on a recent article published by some of the same authors (ref 10, <https://pubmed.ncbi.nlm.nih.gov/37464058/>). The current study found significantly accelerated cognitive declines in AD+LB+ patients compared to the AD+LB- group. Furthermore, the data strengthened the validity of aSyn-SAA as an excellent in vivo biomarker of LB pathology by comparing the SAA results to neuropathological changes in deceased patients from the same cohort. They also confirmed the usage of FDG-PET as a differential diagnostic tool for AD and DLB.

We thank the reviewer for their compliments and appreciation of the work. Please find below a point-by-point response to the suggestions made.

The study is very strong and does not have very significant weaknesses. Perhaps one point not discussed that could be a weakness is the fact that similar to LB pathology, TDP-43 co-pathology has also been reported to be frequent (50-60%) in AD patients and associated with greater cognitive declines (<https://pubmed.ncbi.nlm.nih.gov/24659241/>; <https://pubmed.ncbi.nlm.nih.gov/26224156/>). It was also reported that the TDP-43 pathology could be found in about 53% of post-mortem brains with AD/LB co-pathologies (<https://pubmed.ncbi.nlm.nih.gov/27495267/>). Due to the lack of a validated and reliable in vivo biomarker for TDP-43 pathology, it is not possible at this time to assess whether part of the changes observed in this study could be attributed in part to TDP-43 co-pathology in AD patients. However, I recommend the authors to include TDP-43 staining in the samples with tissue available to analyze whether TDP-43 co-pathology is seen with similar or higher frequency in AD+LB+ and AD+LB- patients. At the very least, they should include as a limitation of this study the putative effect that other co-pathologies (including TDP-43, but maybe also vascular changes) could have in their results.

We thank the reviewer for raising this important point on other common comorbidities, such as TDP-43. We have added a section to the Discussion where this consideration is mentioned:

'Finally, due to the lack of a valid and reliable in-vivo biomarker for TDP-43, this common co-pathology to AD and mixed AD/DLB cases³⁷ was not included in the current work. In line, our post-mortem analysis demonstrated that comorbid TDP-43 pathology (amygdala) was observed in approximately 35% of AD+ cases, respectively. TDP-43 co-pathology has been associated with greater cognitive decline^{38, 39} and therefore of great interest for future studies.'

In addition, we retrieved information on pathological staining of TDP-43 for the hippocampus and amygdala region from the database (NPTDPB and NPTDPC). Figure 1B and Supplementary Table-7 (see below) have been updated accordingly. It can be appreciated that both AD pathological groups had a similar incidence of comorbid TDP-43 pathology in the hippocampus. In addition, TDP-43 pathology was somewhat more prevalent in the negative pathological group. This could explain some of the discrepancies between the clinical diagnosis of MCI or dementia and the absence of AD and LB pathology. However, probably

due to the small numbers, the logistic regression analysis did not result in significant differences between the groups. This has been added to the results section:

'Finally, TDP-43 pathology in the hippocampus and amygdala did not differ between groups (Supplementary Table-6).

	AD-LB- (N=7)	AD-LB+ (N=4)	AD+LB- (N=27)	AD+LB+ (N=23)
Age at death, y	82.6 (6.68)	88.3 (6.80)	81.0 (7.67)	80.5 (7.63)
Sex, F (%)	1 (14.3%)	2 (50.0%)	10 (37.0%)	3 (13.0%)
Measurement interval, y	3.14 (2.85)	5.25 (4.65)	5.22 (3.23)	3.26 (2.20)
Postmortem interval, hours	9.50 (7.35)	13.1 (6.68)	8.88 (8.71)	15.9 (16.4)
Lewy Body pathology				
No	7 (100%)	0 (0%)	20 (74.1%)	1 (4.3%)
Olfactory bulb	0 (0%)	1 (25.0%)	1 (3.7%)	0 (0%)
Brainstem/amygdala predominant	0 (0%)	1 (25.0%)	4 (14.8%)	6 (26.1%)
Cortical	0 (0%)	2 (50.0%)	2 (7.4%)	16 (69.6%)
Thal phases				
0	2 (28.6%)	0 (0%)	0 (0%)	0 (0%)
1	1 (14.3%)	3 (75.0%)	0 (0%)	0 (0%)
2	0 (0%)	0 (0%)	0 (0%)	0 (0%)
3	4 (57.1%)	1 (25.0%)	0 (0%)	1 (4.3%)
4	0 (0%)	0 (0%)	7 (25.9%)	9 (39.1%)
5	0 (0%)	0 (0%)	20 (74.1%)	13 (56.5%)
6	2 (28.6%)	0 (0%)	0 (0%)	0 (0%)
CERAD				
0	4 (57.1%)	3 (75.0%)	1 (3.7%)	2 (8.7%)
1	2 (28.6%)	1 (25.0%)	1 (3.7%)	2 (8.7%)
2	1 (14.3%)	0 (0%)	4 (14.8%)	2 (8.7%)
3	0 (0%)	0 (0%)	21 (77.8%)	17 (73.9%)
Braak stages				
0	1 (14.3%)	0 (0%)	0 (0%)	0 (0%)
1	2 (28.6%)	1 (25.0%)	0 (0%)	0 (0%)
2	2 (28.6%)	3 (75.0%)	1 (3.7%)	3 (13.0%)
3	2 (28.6%)	0 (0%)	1 (3.7%)	0 (0%)
4	0 (0%)	0 (0%)	0 (0%)	1 (4.3%)
5	0 (0%)	0 (0%)	16 (59.3%)	15 (65.2%)
6	0 (0%)	0 (0%)	9 (33.3%)	4 (17.4%)
TDP-43				
Hippocampus (present)	3 (42.9%)	1 (25.0%)	6 (22.2%)	5 (21.7%)
missing	0 (0%)	0 (0%)	0 (0%)	2 (8.7%)
Amygdala (present)	3 (42.9%)	1 (25.0%)	10 (37.0%)	8 (34.8%)
missing	0 (0%)	0 (0%)	0 (0%)	2 (8.7%)

The authors describe that participants with isolated LB pathology were least often APOE-ε4 carriers ($\chi^2=158.98$, $p<0.001$). This is somewhat surprising, considering previous articles showing that the APOE ε4 allele has been associated with an increased risk of DLB and PDD (e.g. <https://www.neurology.org/doi/10.1212/WNL.00000000000006212>). Do the authors have an explanation for these findings?

The reviewer raises an interesting point. While there is indeed some evidence that while APOE-e4 carriership is independently associated with LBD pathology as reported in the referenced paper, the lower prevalence of this genotype in isolated LB+ cases in this cohort was not unexpected. In recent work (doi: <https://doi.org/10.1101/2023.12.05.569878>), our group showed that APOE-e4 carriership in subjects with α-synucleinopathy was only higher in those assigned to the subtype associated with AD comorbidity, while the two more 'pure α - syn' subtypes were associated with a lower prevalence (46% vs. 30/36%). This is in line with the findings of our work. Of note, the ADNI cohort is primarily focused on the AD spectrum, as also reflected in the exclusion criteria. As a result, the isolated LB+ group in our work was small and conclusions should be drawn with caution, as also stated in the limitations section. In line with comments reviewer #4, we added the ADNI exclusion criteria have been added to the Cohort paragraph to illustrate this point:

'Of note, ADNI exclusion criteria includes overt parkinsonian symptoms and neurological dysfunction. As such, patients with PD or DLB are not included in the dataset, which is therefore not representative of such patient populations.'

Reviewer #2 (Remarks to the Author):

Reviewer #3 (Remarks to the Author):

As the authors correctly state, whilst co-pathology with LB in Alzheimer's disease has been well recognised and well studied in neuropathology studies, the lack of a reliable in vivo biomarker for LB pathology has limited addressing this question in clinical cohorts. Seed amplification assays have recently emerged and can finally answer these questions.

The key message of this study is that in those with mild cognitive impairment or Alzheimer's disease, the presence of Lewy body (LB) co-pathology as measured by seed amplification assays (SAA) is associated with worse cognitive performance cross sectionally, and longitudinal cognitive decline. Secondly, they also show increased cortical hypometabolism in posterior brain regions in those with LB co-pathology with FDG PET, and validate the SAA assay in a subgroup with postmortem brain tissue.

The significance of this study is twofold. Firstly, the last two years have seen the first positive clinical trials of disease modifying drugs in mild cognitive impairment (MCI) and early Alzheimer's disease (AD) (lecanemab1 and donanemab2). These treatments both target amyloid pathology, lecanemab being approved by the FDA and donanemab under assessment. It is clear that whilst these drugs show an effect, there are limitations in terms of effect size and safety. This leads to a timely question of other factors and pathologies underlying disease progression in MCI/AD, and the authors quite rightly state that this could

inform stratification for future clinical trials.

Secondly, this study has been made possible by a novel technology to detect Lewy body pathology in vivo in CSF, the SAA. Prior to this studies relied either on post mortem³, α -synuclein measurement in CSF, or using FDG as a proxy for presumed LB pathology⁴. Postmortem studies are thus limited to small sample sizes, FDG PET relies on the assumption of the pattern of hypometabolism being indicative of pathology, and CSF α -synuclein is not a reliable biomarker of LB pathology⁵. With SAA, the idea of measuring the impact of LB-copathology on clinical and imaging metrics has been made possible in a much larger sample size. It also follows that this could be practically carried out either in trials or a clinical setting. The findings are in line with the previous studies using postmortem or other measures, but the demonstration of this effect in a large cohort with a new technology is highly significant and novel, and could have direct implications for clinical trials.

Overall, the data are valid and robust. Firstly, the use of the ADNI cohort, which is a large, well phenotyped cohort, with detailed clinical, biomarker, and imaging data. The novel SAA has been well validated in Lewy body dementias, and detection of LB pathology in other dementias, but the authors take the extra step of demonstrating the validity of the assay within this cohort in the subgroup with postmortem tissue. This identifies similar results to previously published work, suggesting a successful application of this technology to this cohort, and its validity as a marker of LB co-pathology in the larger cohort.

A greater understanding of other features that underly progression in AD beyond amyloid and tau, that can be measured by an in vivo biomarker, could certainly inform future clinical trials that may wish to stratify, test post-hoc, exclude from amyloid targeting trials, or recruit specifically to trials targeting synuclein. LB pathology is commonly found postmortem, but under researched, possibly due to the lack of in vivo biomarkers until recent years. For that reason, I would say this study represents a significant paper for the field. It would also be of interest in the closely related field of dementia with Lewy bodies, and in Parkinson's disease research.

To my knowledge, there are two other studies that explore clinical measures in relation to LB pathology in large neurodegeneration cohorts, including one in ADNI. The recent study in the Biofinder cohort⁶ included 883 individuals, but across diagnosis including LBD. This study identified that the presence of LB pathology, as measured by SAA was associated with attention/executive impairments, visuospatial, motor function, and hallucinations. This study identified more rapid progression independent of diagnostic group.

This submitted paper extends on that, as the AD+/LB- group had 377 people, and the AD+/LB+ group is 98 people. This new study has a similar sized AD group (395), but 177 with AD+/LB+ (dual pathology). This study using the ADNI cohort also differs in that those recruited met criteria for either MCI or AD, so whilst the previous study statistically controlled for the impact of LBD diagnosis on cognitive decline, this study includes larger numbers and only those with an initial AD or MCI diagnosis. This study, as the authors reference, did not detect a significant difference between the AD+ group and the AD+LB+ group in rate of cognitive decline, possibly due to lack a power, which has been solved by this paper. Secondly, the study in the ADNI cohort⁷ showed that LB co-pathology as measured by SAA was associated with cross sectional cognitive impairment in individuals with low tau. This study however did not include data on longitudinal cognitive decline, the key finding of the submitted paper, and whilst included FDG PET did not look at how LB co-pathology impacted FDG PET.

The cohort is a large, well established cohort with rich multimodal data. The choice of CSF results for AD pathology and LB pathology within a year is a reasonable cut off. The standards for AD biomarker positivity are the standards recommended by the central ADNI team, which is appropriate. There is some missing cognitive data, particularly in visuospatial function, with less available at baseline and follow up. Despite this, it remains a large data

set and this missingness does not impact the overall conclusions made by the paper. I have no expertise in the specific FDG PET acquisition and processing, however the methods used are standardised within ADNI. Averaging cortical values across hemispheres is commonly performed and a reasonable assumption based on previous data that bilateral hypometabolism would be found. Similarly, the post mortem assessments are following standard ADNI protocols. The McKeith criteria is a widely used and accepted standard for Lewy body pathology included in international guidelines.

The statistical methods for demographic differences are standard. Using linear mixed effects models with an interaction term between group and time is a commonly used and accepted approach for longitudinal data. They included both random intercepts and slopes, and tested whether a nonlinear model better fit the data, and corrected p values for multiple testing appropriately. Sensible covariates are also included in the model. It is notable the difference between AD+ and AD+LB+ was only significant in the model with the quadratic term, however this model is by far the preferred by BIC, so is the crucial one.

The imaging findings are appropriately corrected for multiple comparisons. Selecting the most different regions at baseline, merging into a meta ROI and then testing for longitudinal associations is reasonable. They also report appropriate correcting for multiple testing and inclusion of covariates. They report the differences in baseline demographics, and age and sex and subsequently included as covariates in analysis.

We thank the reviewer for their thorough review of our work and their appreciation of the manuscript. Please find below a point-by-point response to the specific comments and suggestions.

Suggestions for improvement:

The authors have some interesting thoughts regarding the lack of associations between LB pathology and motor symptoms in this cohort, as ADNI specifically recruited those meeting criteria for AD, or MCI. It would be very interesting to see if this extends to the presence of visual hallucinations. Previous autopsy studies⁸ have highlighted that visual hallucinations are probably the best predictor of LB pathology in mixed AD/LB. Within ADNI itself, visual hallucinations⁹ were associated with similar FDG PET findings found with LB pathology in this paper. It would certainly be of interest clinically whether the novel application of SAAs supported this association with visual hallucinations in this larger cohort. It may also raise the question as to whether the groups with mixed LB and AD pathology, occipital hypoperfusion, and the possible presence of a core feature of DLB could be argued to have DLB with AD co-pathology. This may strengthen the authors argument of the usefulness of SAA in clinical practice or trials, to properly stratify and target interventions. (The ability to test this may rely on specific data requests/access/sharing with the ADNI cohort, and I would say the novelty and significance of the study are clear without this analyses, however it would improve things).

Thank you for this interesting suggestion. Unfortunately, information on the presence of visual hallucinations is limited. From the ADNI database, we retrieved the NPIB variable, which is representative of hallucinations (yes/no) according to the data dictionary (<https://adni.loni.usc.edu/data-dictionary-search/?q=NPIB>) and in line with the paper referenced. Using the same inclusion criteria (CI, within one year of SAA measure) as for the other analysis performed, only 289 cross-sectional observations were available. Of those, only 12 subjects were coded to experience hallucinations. Because of the insufficient statistical

power, we opted not to include this analysis to the manuscript, but rather mention it as a limitation/future direction:

'In addition, these features of parkinsonism and other more LB-pathology associated symptoms such as visual hallucinations might not have been optimally measured in this multicenter study.'

Some have argued that co-incident LB pathology do not impact cognitive decline at the MCI stage¹⁰. It may be of interest if there are sufficient numbers within those with an MCI and AD diagnosis subgroups, whether the effect on cognitive decline is the same or distinct at each disease stage.

We appreciate this valuable comment. While the numbers for the Dementia sub-sample were too small regarding the non-AD pathological groups (16 and 8, see **Table-1**) and follow-up data was limited for the AD+LB- and AD+LB+ groups, an adequate sample size was available for the MCI population.

We performed the linear mixed models for the MMSE, PACC, and the cognitive domains for the MCI and found highly consistent results. Similarly to the whole population, the two AD pathological groups showed the steepest cognitive decline across tests compared to the reference population. In addition, the isolated LB+ group also displays significant worse visuospatial functioning compared to the reference (AD-LB-) population. Please see new **Supplementary Table-4** for detailed results.

Regarding the pairwise comparisons, we observed in agreement with the whole population, that the comorbid pathology group (AD+LB+) demonstrated faster decline on the MMSE and PACC compared to the AD+LB- and additionally AD-LB+ population. In addition, and very interestingly, the isolated LB+ (AD-LB+) group performed worse in visuospatial functioning than the isolated AD (AD+LB-) group. Please see new **Supplementary Table-5** for detailed results. Thus, co-incident LB pathology is also predictive of a worse cognitive outcome at the MCI stage.

Results are further illustrated in new **Supplementary Figure-2** and detailed results have been added to **Supplementary Results**.

Supplementary Figure-2. AD/LB groups and cognitive domains over time in MCI population

Figure illustrates AD/LB group differences for longitudinal cognitive performance based on the LMM. All models were corrected for baseline age, sex, cognitive state, and level of education. The spaghetti plots illustrate raw data regarding cognitive performance over time, while lines represent model fits (shaded area reflect 95% confidence interval). For executive functioning and language domains, a model including an additional quadratic term for time better described the data, while for memory and visuospatial functioning linear models were preferred.

Overall consistency of the results has been added to the main text:

*'LMMs including an additional quadratic term for time ($time^2$) were preferred for the MMSE and PACC based on ΔBIC (Table-2). Over time, both the AD+LB- and AD+LB+ groups showed a significant initial (time) and accelerated ($time^2$) decline compared to the AD- groups (Table-2). The accelerated decline was more pronounced for the AD+LB+ group than the AD+LB- group (MMSE: $\beta_{AD/LB\ group*time^2} = -0.15$, $SE=0.05$, $p=0.005$, PACC: $\beta_{AD/LB\ group*time^2} = -0.026$, $SE=0.008$, $p=0.0015$, Supplementary Table-3), indicating that those with LB pathology in addition to AD pathology accelerated in global cognitive decline over time compared to those with isolated AD pathology (Figure-2C/D). Results were consistent within the MCI population (Supplementary Table-4&5, Supplementary Figure-2A/B).*

For the domain-specific cognitive composites, AD+LB- and AD+LB+ groups performed worse on most cognitive domains compared to the AD-LB- group at baseline and over time. In addition, the AD+LB+ group performed worse than the AD+LB- group for memory, executive functioning, and language, but not visuospatial functioning at baseline. Over time, no cognitive domain-specific differences between AD+LB- and AD+LB+ groups were observed. Results were similar for the MCI population, with the most apparent difference being that the isolated LB+ group performed worse on visuospatial functioning over time compared to both the AD-LB- and AD+LB- groups. Detailed results are described in Supplementary Results.'

For exploratory purposes, we compared the AD+LB- and AD+LB+ groups within the AD dementia subpopulation to investigate whether a similar effect of pathological group on MMSE and PACC performance over time was observed. Based on the quadratic model, we did observe a significant, though marginally, difference between these pathological groups in cognitive decline (MMSE: $\beta = -0.41$, $p=0.03$; PACC: $\beta = -0.06$, $p=0.04$), with the AD+LB+ showing a steeper decline compared to the AD+LB- group (see Figure below). However, probably due to a lack of data, the quadratic model was not preferred. In a linear model, the group comparisons were not significant.

Considering the limited data and resulting lack in statistical power, we opted not to include these results in the main manuscript.

It would be good if there was more explanation of the statistical method of pairwise comparison that led to the results in figure 2c + d, and described in supplementary table 3c. This may well be a standard approach, or I may have missed it in the methods, but it would

help to fully understand this table as it contains what I read as the most important findings of the study.

We thank the reviewer for pointing out this oversight. We used the R package 'emmeans', which is the standard for lme4 package linear mixed models to do the pairwise comparisons. This has now been specified in the methods section.

It is also notable that there are no differences in any individual cognitive domain, although some of these in the table are listed as not available for the quadratic model, whilst not significant in the non quadratic model. Is this because of smaller subsets of data for these domains, and the model not working?

The reviewer is correct that not for all cognitive domains the quadratic model is reported. This is due to the fact that this more complex model, which included an additional term for time but in quadratic in addition to a simpler linear term for time, was not preferred based on the Bayesian Information Criterion (BIC). This criterion provides information on whether additional predictors improve your model fit while giving a penalty for model complexity. It therefore supports the selection of the best and simplest model simultaneously.

In case of the language and visuospatial domains, the more complex model was not preferred as it did not provide sufficient additional variance explained compared to the simpler model. The lack of improved model fit after including a time² term was indeed most likely due to less available data for the visuospatial analysis, which had less subjects, follow-up points, and follow-up time compared to the other domains. For the other domains, it might simply be that a more linear decline is observed. For example, even with the quadratic model preferred for the memory (**Sup figure 1E**) and executive functioning (**Sup figure 1F**) domain, one can appreciate that the trajectory for the AD+LB- and AD+LB+ groups is quite linear, and rather the other two groups demonstrate a non-linear trajectory.

Nonetheless, it is important to point out that several group differences were apparent for the cognitive domains in an expected direction, with the AD+LB+ group performing worse at baseline compared to the AD group for all domains, except for memory for example.

The changes in regional metabolism between AD+ and AD+LB+ groups are well described and the figures present this clearly. The longitudinal change is reported to be at the statistical trend level of 0.06. The limitations of p-values and arbitrary nature of a 0.05 cut off present limitations, but for better or worse it remains a standard. Whilst in the results section and figure this is presented, the figure description puts this as 'more apparent hypometabolism'. It might be simpler to describe this as a non significant trend towards longitudinal reductions in metabolism or similar.

We agree with this comment. The figure text has therefore been adapted to:

*'C) The spaghetti plots illustrate raw data regarding FDG-PET over time, while lines represent model fits (shaded area reflect 95% confidence interval), demonstrating a **non-significant trend towards** more apparent hypometabolism **over time** in the meta-ROI, consisting of regions identified in B.'*

Line 224 may need a bit more discussion/clarification – would describing the pathology as more widespread or contrasting the cortical with brainstem/amygdala help? It is clear what is meant from looking at the supplementary table but I think a little more in the text would be useful.

We thank the reviewer for pointing this out. We adapted the text to clarify the difference between more widespread and early LB pathology:

*'Among the cases with cortical LBs **indicative of widespread pathology**, 90% (18/20) had a positive a-synuclein SAA test. In turn, only 63.6% (7/11) of those with **relatively early LB pathology** localized predominantly in the brainstem or amygdala had a positive SAA test. Among those with no LB pathology, 96.4% (27/28) had a negative SAA test result, illustrating the tests excellent specificity (Figure-1B).'*

I believe the manuscript references the previous literature appropriately. It may be worth mentioning the Landau reference⁷ as it uses the SAA in the same cohort, although may well have been published after submission.

Indeed, this paper was published while the current work was under review. It has now been added to the Discussion, where the importance of comorbidity in the context of trials is discussed:

*'Importantly, our neuropathological results suggest that this is not merely due to more severe AD pathology in the mixed pathology group, but rather the additive effect of concomitant LB pathology. **In line, recent work in the same cohort suggests that α -synuclein pathology was associated with poorer cognition particularly when tau was low²⁴.***

We additionally referenced this recent publication in the discussion: <https://alz-journals.onlinelibrary.wiley.com/doi/full/10.1002/alz.13818>

1. Van Dyck, C. H. et al. Lecanemab in Early Alzheimer's Disease. N. Engl. J. Med. 388, 9–21 (2023).
2. Sims, J. R. et al. Donanemab in Early Symptomatic Alzheimer Disease: The TRAILBLAZER-ALZ 2 Randomized Clinical Trial. JAMA 330, 512 (2023).
3. Toledo, J. B. et al. Clinical and multimodal biomarker correlates of ADNI neuropathological findings. Acta Neuropathol. Commun. 1, 65 (2013).
4. Silva-Rodríguez, J. et al. Characteristics of amnesic patients with hypometabolism patterns suggestive of Lewy body pathology. Brain 146, 4520–4531 (2023).
5. Gao, L. et al. Cerebrospinal fluid alpha-synuclein as a biomarker for Parkinson's disease diagnosis: a systematic review and meta-analysis. Int. J. Neurosci. 125, 645–654 (2015).
6. Quadalti, C. et al. Clinical effects of Lewy body pathology in cognitively impaired individuals. Nat. Med. 29, 1964–1970 (2023).
7. Landau, S. M. et al. Individuals with Alzheimer's disease and low tau burden: Characteristics and implications. Alzheimers Dement. 20, 2113–2127 (2024).
8. Thomas, A. J. et al. Improving the identification of dementia with Lewy bodies in the context of an Alzheimer's-type dementia. Alzheimers Res. Ther. 10, 27 (2018).
9. for the Alzheimer's Disease Neuroimaging Initiative, Pezzoli, S., Manca, R., Cagnin, A. & Venneri, A. A Multimodal Neuroimaging and Neuropsychological Study of Visual Hallucinations in Alzheimer's Disease. J. Alzheimers Dis. 89, 133–149 (2022).
10. Shriram, J., Malek-Ahmadi, M., Irwin, C. & Sabbagh, M. Impact of incidental synucleinopathy in mild cognitive impairment due to Alzheimer disease. J. Neuropathol. Exp. Neurol. 83, 230–237 (2024).

Reviewer #4 (Remarks to the Author):

This is a study of the additional detrimental effects of α -synuclein pathology in a large well-established cohort of mild cognitive impaired and demented individuals with presumed

Alzheimer's disease. The methods are relevant and clearly described, figures easy to understand and results interpreted with respect to limitations. Overall the study is highly relevant and provide original and new insight to a major research field in neuroscience using a rather new but established method for measuring α -synuclein in CSF. In a subset of the patients, the method is further validated to post-mortem assessments.

I have only encountered a few issues that should be dealt with.

We thank the reviewer for the appreciation of our work and a point-by-point response to the provided suggestions can be found below.

In the methods section, it says that participants with a CSF α -synuclein SAA test were selected. As this measure is not part of the standard test for ADNI subjects according to adni.loni.usc.edu, please describe in more detail how the subjects were selected. From which centers were they included and which inclusion criteria? Were any CSF α -synuclein SAA test performed retrospectively on biobank material? And if so how were the subjects selected? Detailed description of the selection of the included subjects is mandatory for evaluating the transferability of results to other patient groups.

The reviewer is correct that the CSF SAA assay is not a standard test within the ADNI cohort but has been recently added to the database by the consortium. The test was performed centrally by the Amprion Clinical Laboratory on the most recent CSF sample available for the ADNI subjects, so indeed retrospectively on biobank material. Therefore, the selection of cases was carried out by the ADNI team. All relevant documentation can be found in the ADNI data dictionary and in Arnold et al., (2022) as referenced in the work.

In the data dictionary, the following selection criteria are stated:

'Selection criteria included using the last CSF time-point available from all ADNI participants. A small number of specimens (n=17) with reported results were visibly discolored. Blood/hemoglobin contamination at levels causing visible discoloration may cause depression of amplification signal in this assay. Therefore, results for visibly discolored specimens (flagged in results table) should be interpreted with caution.'

Thus, for the current work, we did not perform any additional selection, rather we included everyone with available CSF SAA, only excluding cases with an intermediate result (i.e. not reliable status) and if the AD biomarkers ($A\beta_{42}$ and $p\text{-tau}_{181}$) were not available within one year of the SAA measure.

This information has been further clarified in the Methods section Study cohort:

'We selected all cognitively impaired participants for which α -synuclein status was determined using the CSF α -synuclein SAA test (using the last available CSF time-point) and who had available data on CSF $A\beta_{42}$ and $p\text{-tau}_{181}$ within one year of α -synuclein SAA status. Cases with an intermediate SAA result were excluded (n=17).'

We have reached out to the ADNI biomarker core lead for further information on the selection procedure, if any beyond simple availability of CSF samples. We did not yet receive a response, but any additional information will be added to the manuscript if received and needed.

40.8% were female. This seems like a low fraction. Do you have any explanation for this?

The reviewer is correct that this percentage of females is somewhat lower than expected for an AD study or scientific studies in general, where an overrepresentation of females is often reported. This might be a chance effect due to the availability of CSF samples (see response above), as no additional data selection has been carried out for the current work, and previous studies reporting ADNI demographics generally describe a higher percentage of females. Still, we observe a tendency for higher percentage of males in isolated LB+ group, in line with literature (<https://doi.org/10.1016/j.parkreldis.2023.105285>).

Patients are included as MCI or AD. Thus, patients with known LBD or PD will be excluded and LB pathology will be a random finding. Therefore, the AD-LB+ group is not representative for patients with symptoms from LB pathology. This limitation is stated briefly in the discussion but could be included in the description of the study cohort as well.

We agree with the reviewer that the ADNI cohort is indeed a cohort that primarily focused on the AD spectrum, resulting in the small group of isolated LB-pathology individuals (AD-LB+). The ADNI exclusion criteria have been added to the Cohort paragraph to illustrate this point:

‘Of note, ADNI exclusion criteria includes overt parkinsonian symptoms and neurological dysfunction. As such, patients with PD or DLB are not included in the dataset, which is therefore not representative of such patient populations.’

The three figures are nicely performed but I suggest to include of number of individuals in each group at baseline and number of individuals with follow-up data in the legends or figures to increase the readability.

We appreciate the reviewers suggestion. We experimented with this suggestion but came to the conclusion that it decreased the readability of the figure. Alternatively, we included a supplementary table with exact N and observations per pathology group and cognitive outcome measure. Baseline numbers can be retrieved from **Table-1**.

This table is now reference in the Methods section, ‘Neurological and neuropsychological assessments’:

*‘Detailed longitudinal numbers per pathological group and cognitive outcomes can be found in **Supplementary Table-7**.’*

Supplementary Table-7. Number of cases and observations for linear mixed model

	AD-LB-	AD-LB+	AD+LB-	AD+LB+
MMSE	N=164 898 obs	N=36 189 obs	N=350 1484 obs	N=160 590 obs
PACC	N=164 899 obs	N=36 190 obs	N=350 1489 obs	N=160 589 obs
Memory	N=164 856 obs	N=36 182 obs	N=350 1484 obs	N=161 590 obs
Executive functioning	N=164 852 obs	N=36 181 obs	N=349 1469 obs	N=159 576 obs
Language	N=164 853 obs	N=36 181 obs	N=350 1478 obs	N=161 588 obs
Visuospatial	N=103 416 obs	N=25 97 obs	N=266 1026 obs	N=137 450 obs

REVIEWERS' COMMENTS

Reviewer #1 (Remarks to the Author):

The authors did a good job answering my concerns. I have not further recommendations.

Reviewer #2 (Remarks to the Author):

Reviewer #3 (Remarks to the Author):

I am grateful for the responses to the comments by the authors. As they report, the limitations on the data collected on visual hallucinations in ADNI would lead to underpowered comparisons, so this it is the right approach to not include this, and I appreciate them looking into the data.

It is very interesting to see that the isolated LB group had worse visuospatial performance at the MCI stage, thank you for completing this additional analysis.

Thank you for adding this detail regarding the statistical analysis.

Thank you for this clear explanation

Thank you - this wording 'non significant trend towards' I think captures the important potential finding whilst being clear about the exact results.

Thank you for adding this clarifying statement

Thank you for including those references.

Reviewer #4 (Remarks to the Author):

This is indeed a highly relevant and sound study on the effects of α -synuclein pathology in Alzheimer patients. The effects on cognitive function and regional cortical metabolism is very convincing. Further, the authors have dealt with the minor issues identified.